# Mustard Gold in the Oleninskoe Gold Deposit, Kolmozero–Voronya Greenstone Belt, Kola Peninsula, Russia

**Arkadii A. Kalinin \*, Yevgeny E. Savchenko and Ekaterina A. Selivanova**

Geological Institute, Kola Science Center, Russian Academy of Science, 184200 Apatity, Russia; evsav@geoksc.apatity.ru (Y.E.S.); selivanova@geoksc.apatity.ru (E.A.S.)
* Correspondence: kalinin@geoksc.apatity.ru; Tel.: +7-921-663-68-36

**Abstract:** The Oleninskoe intrusion-related gold–silver deposit is the first deposit in the Precambrian of the Fennoscandian Shield, where mustard gold has been identified. The mustard gold replaces küstelite with impurities of Sb and, probably, gold-bearing dyscrasite and aurostibite. The mosaic structure of the mustard gold grains is due to different orientations and sizes of pores in the matrix of noble metals. Zonation in the mustard gold grains is connected with mobilization and partial removal of silver from küstelite, corresponding enrichment of the residual matter in gold, and also with the change in the composition of the substance filling the pores. Micropores in the mustard gold are filled with iron, antimony or thallium oxides, silver chlorides, bromides, and sulfides. The formation of mustard gold with chlorides and bromides shows that halogens played an important role in the remobilization of noble metals at the stage of hypergene transformation of the Oleninskoe deposit.

**Keywords:** Fennoscandian Shield; Kolmozero–Voronya belt; Oleninskoe deposit; mustard gold; dyscrasite; chlorargyrite

## 1. Introduction

Mustard gold (the term is used below regardless of the Au/Ag ratio in it, and terms "electrum" for Au/Ag = 1/1 and "küstelite" for Au/Ag = 1/3 are used when the composition of Au–Ag alloy is important) is a relatively rare mineral aggregate, formed in the zone of hypergenesis as a result of oxidation and decomposition of gold tellurides and antimonides. It represents a lacy network, or sponge of native gold with very fine (often less than a micrometer) pores either filled with Fe, Te, Pb, Cu, Au, Ag, Sb, Hg oxides or unfilled. These aggregates were named mustard gold because of their distinctive appearance—loose texture and yellow–brown to brown color caused by Fe-oxides filling the pores. Mustard gold has low reflectivity due to high porosity.

The history of the study of mustard gold is not long. The term was first used in geological literature, following a traditional miners' lexicon, by Waldemar Lindgren in 1933 [1] in the paragraph about the oxidation zone in Au–Te deposits. A more extensive study of mustard gold refers to the end of the XX and the beginning of the XXI century—to the time of fast development of local methods of mineral analysis.

Mustard gold is often found in oxidized ores of Au–Ag–Sb and Au–Ag–Te deposits [2], mainly in weathered quartz granular rocks in the zone of secondary sulfide enrichment. Individual mustard gold grains are rare. More often, mustard gold forms aggregates with native gold: It can grow on the surface or inside gold grains, or be overgrown by late high-grade gold [3].

The composition of mustard gold can vary within an individual deposit. For example, eight varieties were determined in the Dongping gold deposit, where 30–50% of Au is concentrated in

the mustard gold [4]. The mustard gold grains in the Dongping deposit differ in the composition of gold, which makes the sponge—it can be high-grade gold and/or gold alloys with lead and with silver. The pores are filled either by goethite, or various tellurites and tellurates or remain unfilled [4]. Six varieties of mustard gold are described in Sb–Au deposits in Sakha–Yakutia [5].

The structure of individual mustard gold grains is often heterogeneous. Many grains are zonal or concentrically banded, other display spotted, or mosaic textures [6,7]. The heterogeneity appears due to gradual decomposition and re-deposition of the material.

As the size of the pores is usually less than 1 micrometer, and thickness of gold filaments ranges from 200 to 500 nm, the microprobe analysis of the mustard gold gives some summary results, averaging data on gold filaments and the substance filling the pore, and shows a presence of Au, Ag, Fe, Te, Pb, Cu, Sb, Hg, etc. It can be difficult to define mineral phases correctly, and microprobe analysis appears rather qualitative than quantitative. Total estimates from the microprobe analyses are often significantly less than 100%. The deficit appears due to unfilled pore space and the presence of oxides.

Mustard gold was found and studied in the deposits in the Far East in Russia (in the Aginskoe, Ozernovskoe, Asachinskoe deposits and Gaching ore occurrence in the Kamchatka Peninsula [6–9], in the Nizhne–Myakitskoe ore field in the Magadan Region [3], in the Tumannoe deposit in Chukotka [10], in the Kuranah ore field, and in the Sarylahskoe and Sentachanskoe gold–antimony deposits in Sakha–Yakutia [9,11]), in China (the Dongping [4,12] and Sandaowanzi [12], gold deposits), in Slovakia (the Kriváň deposit in the High Tatra Mts.) [13], and in Bolivia (Au–Sb deposit Kharma) [14]. All these deposits are located in the areas of Mesozoic–Cenozoic volcanism with numerous epithermal Au–Te and Au–Sb gold deposits. Mustard gold was not described earlier in the Precambrian gold deposits, and, probably, our finding in the Oleninskoe deposit is the first such occurrence described.

## 2. Geological Setting of the Oleninskoe Deposit

The Oleninskoe is a small gold deposit (~10 t or 0.32 Moz @ 7.6 g/t Au [15,16]), located in the Neoarchean Kolmozero–Voronya greenstone belt, which separates two major tectonic units of the Fennoscandian Shield—the Murmansk craton and the Kola Province (Figure 1A). The geological position and structure of the Oleninskoe deposit are described in [15–18], and here is a summary of the published data.

The location of the deposit is controlled by a shear zone of the NW strike in the amphibolite of the Oleny Ridge sequence (Figure 1) [15,16]. The amphibolite and high-alumina metasedimentary schist in the area of the deposit contain numerous granodiorite quartz porphyry sills 0.1–6.0 m thick (Figure 1). The rocks were low amphibolite (T = 600 °C, P = 3–4 kbar) metamorphosed in Neoarchean. The youngest rocks (2.45 Ga [19]) in the deposit area are granite–pegmatite veins, which cut all metamorphic rocks, including those hosting gold mineralization (Figure 1).

Amphibolite and granodiorite porphyry are intensely altered, the alteration zone is ~50 m thick and traced along the strike for 200–250 m. The whole zone of alteration is covered with early biotitization (potassium metasomatism) and formation of diopside–zoisite–carbonate mineral assemblage (calcium metasomatism) in the amphibolite [15,16]. Quartz-rich metasomatic rocks (quartz–muscovite–albite, quartz–tourmaline, and quartz rocks) formed later, replacing amphibolite and granodiorite porphyry as an echelon-like series of three lens bodies, cutting general schistosity in the host rocks at an acute angle of 10–15° (Figure 1): We interpret the structure of the deposit [15] as a strike-slip fault bridge structure (this kind of structures is described in [20]). The lenses are up to 3.5 m thick (1.5 m on the average) with the length up to 50 m. These quartz-rich rocks control the distribution of the gold–silver mineralization.

The thickness of the Quaternary glacial sediments is 0–2 m. The mineralized lenses are exposed with a number of trenches and drill holes (Figure 1). The zone of ore oxidation is less than 1 m thick.

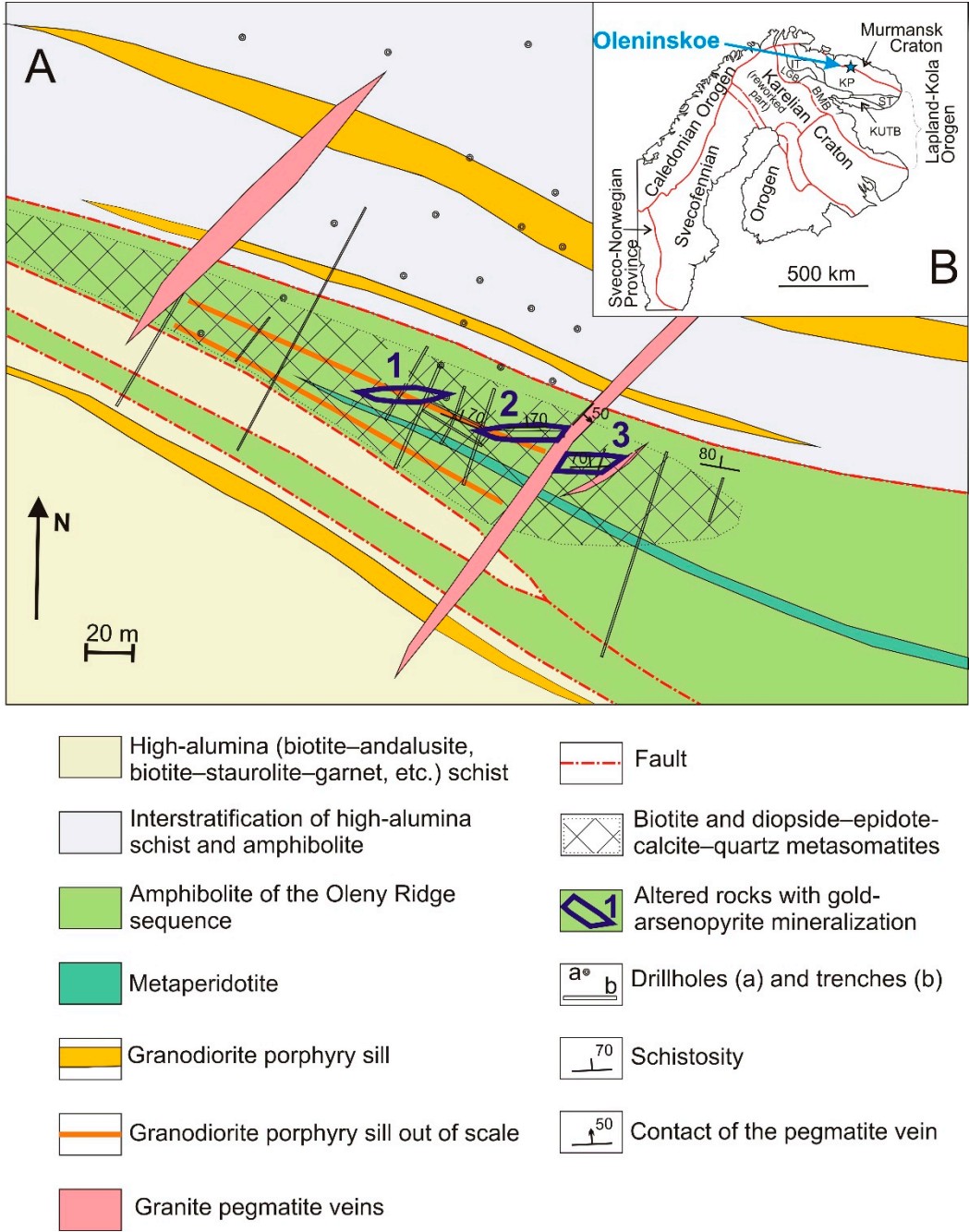

**Figure 1.** Schematic geological map of the Oleninskoe gold deposit (**A**) and the position of the Oleninskoe deposit in the tectonic map of the Fennoscandian Shield (**B**). KP—Kola province; BMB—Belomorian mobile belt; LGB—Lapland granulite belt; KUTB—Kolvitsa–Umba–Tersk belt; IT—Inari terrane; ST—Strel'na terrane.

Arsenopyrite, pyrrhotite, and ilmenite are the most abundant ore minerals, which are present in all altered rocks. Quartz–tourmaline and quartz metasomatic rocks in lens #2 (Figure 1) contain rich Pb–Ag–Sb mineralization (galena, freibergite, dyscrasite, boulangerite, semseyite, diaphorite, pyrargyrite, and other Sb-sulfosalts of Pb, Ag, and Cu—totaling more than 40 mineral species [21]).

The age of mineralization is probably Neoarchean because mineralized rocks are cut by 2.45 Ga pegmatite veins, although Volkov and Novikov [17] reported some signatures of overprinting hydrothermal events of Paleoproterozoic age (1.9 Ga) in the deposit. The temperature of the formation of mineralization was estimated at 550–300 °C [18,22] with arsenopyrite geothermometer.

Four types (generations) of minerals from the Au–Ag series were recognized in the deposit [21]: (1) küstelite 25–32 wt% Au in association with arsenopyrite, löllingite, and pyrrhotite; (2) electrum 33–47 wt% Au in intergrowths with galena, dyscrasite, and sulfosalts; (3) gold (78–95 wt% Au) in quartz; (4) native silver (<7 wt% Au). Types 1 and 3 are widespread in the deposit, 2 and 4 relate only to the quartz and quartz–tourmaline rocks rich in Pb–Ag–Sb. Other gold minerals in the Oleninskoe deposit are gold-bearing dyscrasite (grains up to 1 mm), aurostibite, petzite, calaverite (inclusions <10 μm in electrum of the 2nd type), and yutenbogaardtite. The latter forms rims around the grains of küstelite and electrum. The gold content in dyscrasite varies from 0 to 17 wt.%, some grains are zonal with the outer parts enriched in S and poor in Au.

The genetic type of the Oleninskoe deposit is debatable. It was classified earlier as a greenstone orogenic deposit [23], but its geochemical characteristics, mineral composition of the ore [15,21], and the composition of fluid inclusions [24] correspond better to the class of intrusion-related deposits.

## 3. Materials and Methods

Gold and silver mineralization was studied in the specimens, collected by the authors in 1981–1983 and again in 2017. The collection included specimens of metasomatic rocks, formed after amphibolite and quartz porphyry, and heavy mineral concentrates from crushed samples of the mineralized rocks and from the glacial deposits, overlaying the ore bodies. Polished sections of mineralized specimens and of heavy mineral concentrates were studied with reflected light microscopy (optical microscope Axioplane) in the Geological Institute of the Kola Science Center.

Preliminary estimation of composition of mineral species, the study of element distribution and intraphase heterogeneity were conducted with LEO-1450 scanning electron microscope (SEM) (Carl Zeiss, Oberkochen, Germany) equipped with a Bruker XFlash-5010 Nano GmbH (Bruker, Bremen, Germany) energy-dispersive spectrometer (EDS). Microprobe analyses were performed for grains larger than 20 μm with MS-46 CAMECA, operating at an accelerating voltage 22 kV, beam current 30–40 nA. The following standards and analytical lines were used: $Fe_{10}S_{11}$ (FeK$\alpha$, SK$\alpha$), $Bi_2Se_3$ (BiM$\alpha$, SeK$\alpha$), $LiNd(MoO_4)_2$ (MoL$\alpha$), Co (CoK$\alpha$), Ni (NiK$\alpha$), Pd (PdL$\alpha$), Ag (AgL$\alpha$), Te (TeL$\alpha$), Au (AuL$\alpha$).

The specific features of analysis of mustard gold, described in the Introduction, made it possible to characterize the chemical composition of the mustard gold not only with the microprobe analysis but with qualitative data obtained using an energy-dispersive system Bruker XFlash-5010 (Tables 1 and 2).

Visually homogenous material from the areas close to the points of microprobe analysis, 50 × 10 μm or more in size, was extracted from the polished sections and examined with the X-ray powder diffraction (Debye–Scherer) on URS-1 operated at 40 kV and 16 mA with RKU-114.7 mm camera and FeK$\alpha$-radiation to identify mineral phases making the mustard gold grains.

**Table 1.** Estimation of composition of mustard gold in weight percent with an energy-dispersive spectrometer.

| Point # | 1 | 2 | 3 | 4 | 5 | 6 | 7 | 8 | 9 | Point # | 10 | 11 | 12 | 13 | 14 | 15 | 16 | 17 | 18 | 19 |
|---|---|---|---|---|---|---|---|---|---|---|---|---|---|---|---|---|---|---|---|---|
| Mg | n.a. | n.a. | 0.18 | n.a. | n.a. | n.a. | n.a. | n.a. | n.a. | S | 0.42 | 7.67 | 1.33 | 1.07 | 1.28 | 0.71 | 8.84 | 4.52 | 0.48 | 0.53 |
| Si | n.a. | n.a. | 1.18 | n.a. | n.a. | n.a. | n.a. | n.a. | n.a. | Cl | 9.23 | 0.09 | 0.36 | 0.13 | 0.32 | 0.13 | 0.15 | 0.11 | 11.33 | 0.09 |
| S | n.a. | n.a. | 0.99 | n.a. | 0.44 | 0.76 | n.a. | 0.61 | 0.75 | Fe | 0.14 | 0.62 | n.a. | n.a. | n.a. | 0.47 | 1.33 | 0.68 | n.a. | n.a. |
| Cl | n.a. | n.a. | n.a. | n.a. | 0.05 | 0.08 | n.a. | 0.18 | 0.13 | As | n.a. | 1.22 | n.a. | n.a. | n.a. | 0.26 | 0.75 | 0.31 | n.a. | n.a. |
| Fe | n.a. | n.a. | 18.76 | n.a. | 0.34 | 3.20 | n.a. | 0.42 | 5.43 | Br | 0.59 | 0.41 | 3.29 | 5.74 | 2.13 | 1.40 | 1.19 | 1.78 | 0.67 | 14.56 |
| As | n.a. | n.a. | n.a. | n.a. | 0.21 | 2.00 | n.a. | 0.21 | 1.19 | Ag | 46.50 | 44.35 | 43.84 | 50.85 | 28.89 | 24.49 | 54.76 | 26.03 | 46.53 | 43.12 |
| Br | n.a. | n.a. | n.a. | n.a. | 2.46 | 3.42 | n.a. | 1.41 | 1.37 | Sb | n.a. | n.a. | 1.79 | 1.00 | 0.66 | 0.50 | n.a. | n.a. | n.a. | n.a. |
| Ag | 58.53 | 51.17 | 7.60 | 63.59 | 52.67 | 33.00 | 64.88 | 46.43 | 35.74 | Au | 30.24 | 31.54 | 43.72 | 28.44 | 54.30 | 59.77 | 22.42 | 53.02 | 29.16 | 30.90 |
| Sb | 2.60 | 1.60 | 17.23 | 2.30 | 1.87 | 0.76 | 1.68 | 0.85 | 0.88 | C | n.a. | n.a. | n.a. | n.a. | 3.64 | n.a. | n.a. | n.a. | n.a. | n.a. |
| Au | 27.84 | 35.32 | 21.97 | 23.76 | 33.46 | 47.97 | 25.73 | 40.35 | 45.17 | Total | 87.12 | 85.90 | 94.33 | 87.23 | 91.22 | 87.73 | 89.44 | 86.45 | 88.17 | 89.20 |
| Pb | n.a. | n.a. | 3.28 | n.a. | n.a. | n.a. | n.a. | n.a. | n.a. | | | | | | | | | | | |
| Total | 88.97 | 88.09 | 71.29 * | 89.65 | 91.50 | 91.19 | 92.29 | 90.46 | 90.66 | | | | | | | | | | | |

| | | | | | | | | | Atomic quantities (%) | | | | | | | | | | | |
|---|---|---|---|---|---|---|---|---|---|---|---|---|---|---|---|---|---|---|---|---|
| Point # | 1 | 2 | 3 | 4 | 5 | 6 | 7 | 8 | 9 | Point # | 10 | 11 | 12 | 13 | 14 | 15 | 16 | 17 | 18 | 19 |
| Mg | n.a. | n.a. | 1.0 | n.a. | n.a. | n.a. | n.a. | n.a. | n.a. | S | 1.5 | 28.3 | 5.6 | 4.6 | 4.3 | 3.8 | 29.0 | 20.3 | 1.6 | 2.2 |
| Si | n.a. | n.a. | 5.5 | n.a. | n.a. | n.a. | n.a. | n.a. | n.a. | Cl | 30.0 | 0.3 | 1.4 | 0.5 | 1.0 | 0.6 | 0.4 | 0.4 | 34.7 | 0.3 |
| S | n.a. | n.a. | 4.1 | n.a. | 1.9 | 3.3 | n.a. | 2.7 | 3.2 | Fe | 0.3 | 1.3 | n.a. | n.a. | n.a. | 1.4 | 2.5 | 1.8 | n.a. | n.a. |
| Cl | n.a. | n.a. | n.a. | n.a. | 0.2 | 0.3 | n.a. | 0.7 | 0.5 | As | n.a. | 1.9 | n.a. | n.a. | n.a. | 0.6 | 1.1 | 0.6 | n.a. | n.a. |
| Fe | n.a. | n.a. | 44.2 | n.a. | 0.8 | 8.1 | n.a. | 1.1 | 13.4 | Br | 0.9 | 0.6 | 5.6 | 9.8 | 2.9 | 3.0 | 1.6 | 3.2 | 0.9 | 24.0 |
| As | n.a. | n.a. | n.a. | n.a. | 0.4 | 3.8 | n.a. | 0.4 | 2.2 | Ag | 49.7 | 48.6 | 55.2 | 64.3 | 28.9 | 38.5 | 53.4 | 34.8 | 46.8 | 52.7 |
| Br | n.a. | n.a. | n.a. | n.a. | 4.2 | 6.0 | n.a. | 2.5 | 2.4 | Sb | n.a. | n.a. | 2.0 | 1.1 | 0.6 | 0.7 | n.a. | n.a. | n.a. | n.a. |
| Ag | 76.9 | 71.1 | 9.3 | 80.9 | 67.0 | 43.2 | 80.6 | 62.0 | 45.7 | Au | 17.7 | 18.9 | 30.2 | 19.7 | 29.7 | 51.4 | 12.0 | 38.8 | 16.1 | 20.7 |
| Sb | 3.0 | 2.0 | 18.6 | 2.6 | 2.1 | 0.9 | 1.8 | 1.0 | 1.0 | C | n.a. | n.a. | n.a. | n.a. | 32.7 | n.a. | n.a. | n.a. | n.a. | n.a. |
| Au | 20.0 | 26.9 | 14.7 | 16.5 | 23.3 | 34.4 | 17.5 | 29.5 | 31.6 | | | | | | | | | | | |
| Pb | n.a. | n.a. | 2.1 | n.a. | n.a. | n.a. | n.a. | n.a. | n.a. | | | | | | | | | | | |

Note: *—includes Al 0.10 wt. %; n.a.—not analyzed.

**Table 2.** Electron microprobe data in weight percent for the mustard gold.

| Point # | 20 | 21 | 22 | 23 | 24 | 25 | 26 | 27 | 28 | 29 | 30 |
|---|---|---|---|---|---|---|---|---|---|---|---|
| S | n.a. | n.a. | n.a. | n.a. | 0.27 | 0.24 | 0.22 | 0.21 | 0.23 | 0.04 | 11.15 |
| Cl | n.a. | n.a. | n.a. | n.a. | 11.16 | 11.44 | 11.49 | 11.48 | 11.61 | 10.72 | 0.00 |
| Fe | 13.5 | 11.16 | 1.71 | 2.08 | 0.04 | 0.10 | 0.14 | 0.00 | 0.05 | 0.00 | 0.15 |
| Al | 0.19 | 0.21 | n.a. | n.a. | n.a. | n.a. | n.a. | n.a. | n.a. | n.a. | n.a. |
| Cu | n.a. | n.a. | 0.25 | 0.24 | n.a. | n.a. | n.a. | n.a. | n.a. | n.a. | n.a. |
| As | 1.16 | 1.26 | 0.61 | 0.74 | 0.13 | 0.00 | 0.00 | 0.00 | 0.00 | 0.00 | 0.00 |
| Ag | 0.81 | 0.52 | 12.22 | 10.38 | 49.90 | 47.38 | 49.15 | 48.11 | 47.37 | 81.49 | 59.86 |
| Sb | 0.11 | 0.18 | 0.42 | 0.32 | n.a. | n.a. | n.a. | n.a. | n.a. | n.a. | n.a. |
| Au | 66.27 | 72.33 | 73.19 | 65.88 | 38.51 | 36.45 | 35.72 | 37.39 | 36.25 | 1.20 | 26.88 |
| Tl | n.a. | n.a. | 5.02 | 6.34 | n.a. | n.a. | n.a. | n.a. | n.a. | n.a. | n.a. |
| Total | 82.04 | 85.66 | 93.43 | 85.97 | 100.00 | 95.60 | 96.72 | 97.20 | 95.51 | 93.45 | 98.04 |
| Atomic quantities (%) | | | | | | | | | | | |
| S | n.a. | n.a. | n.a. | n.a. | 0.8 | 0.8 | 6.7 | 0.7 | 0.8 | 0.1 | 33.4 |
| Cl | n.a. | n.a. | n.a. | n.a. | 32.0 | 33.7 | 31.4 | 33.5 | 34.2 | 28.4 | 0.0 |
| Fe | 40.2 | 25.3 | 5.5 | 7.2 | 0.1 | 0.2 | 0.2 | 0.0 | 0.1 | 0.0 | 0.3 |
| Al | 1.2 | 1.0 | n.a. | n.a. | n.a. | n.a. | n.a. | n.a. | n.a. | n.a. | n.a. |
| Cu | n.a. | n.a. | 0.7 | 0.7 | n.a. | n.a. | n.a. | n.a. | n.a. | n.a. | n.a. |
| As | 2.6 | 25.3 | 1.5 | 1.9 | 0.2 | 0.0 | 0.0 | 0.0 | 0.0 | 0.0 | 0.0 |
| Ag | 1.2 | 2.1 | 20.4 | 18.7 | 47.0 | 45.9 | 44.1 | 46.2 | 45.8 | 70.9 | 53.3 |
| Sb | 0.2 | 0.6 | 0.6 | 0.5 | n.a. | n.a. | n.a. | n.a. | n.a. | n.a. | n.a. |
| Au | 55.9 | 46.6 | 66.9 | 64.9 | 19.9 | 19.4 | 17.5 | 19.7 | 19.2 | 0.6 | 13.1 |
| Tl | n.a. | n.a. | 4.4 | 6.0 | n.a. | n.a. | n.a. | n.a. | n.a. | n.a. | n.a. |

Note: n.a.—not analyzed.

## 4. Results

Mustard gold was found in quartz-rich metasomatic rocks with rich Au–Ag–Sb–Pb mineralization in lens #2 (Figure 1) and in the overlaying glacial deposits. The first grain of mustard gold forms an intergrowth with küstelite (Au 28–35 wt%); the mustard gold comprises the central part of the grain, but is overgrown by küstelite, and forms a thin rim around the küstelite (Figure 2).

The mustard gold is brown-gray under reflected light, strongly anisotropic, with a yellow color effect under crossed polars. The mosaic structure of this mustard gold can be easily seen even with an optical microscope (Figure 2A), and with an SEM we see that the blocks differ in size of pores, which varies from the nanoscale to 8 μm, and in the direction of the gold filaments (Figure 2C–E). This mustard gold contains Au, Fe, Sb, Ag, Pb, and O (17.0 wt%) as the main components, and minor S, Si, Al, Mg (Table 1, #3). X-ray analysis of the mustard gold shows the presence of Ag–Au alloy, but Fe and Sb oxides are not identified.

Another grain of mustard gold ~0.1 mm in size was noted in association with electrum (wt%: Au 61.2%, Ag 34.6%, Sb 0.5%, total 96.3%). The grain is brick red, strongly anisotropic, with a bright yellow color effect in reflected light under crossed polars. This grain exhibits block structure, the size of the particles is 0.1–0.3 μm. Microprobe analysis shows that the main elements are Au and Fe, and minor elements (<1 wt%) are As, Ag, Sb, and Al (Table 2, #20 and 21). The low "Total" values are due to porosity and oxides. This mustard gold grain is coated with a thin rim of sulfur-rich simplesite (wt%: Fe 30.62%, As 15.11%, S 4.24%, Sb 0.10%) 5–10 μm thick.

One of the studied grains of mustard gold contains 5–6 wt% of Tl (Table 2, #22, and #23). This grain has brick red color, and its appearance is very similar to native copper (Figure 3). The backscatter electron image shows fine intergrowth of different mineral phases. Except for Au, Ag, and Tl, the substance contains Fe (~2%), minor As, Cu, Sb (Table 2). Zonation in the grain reflects the change in gold, silver, iron, and thallium content. The low "Total" values are due to oxides and porosity.

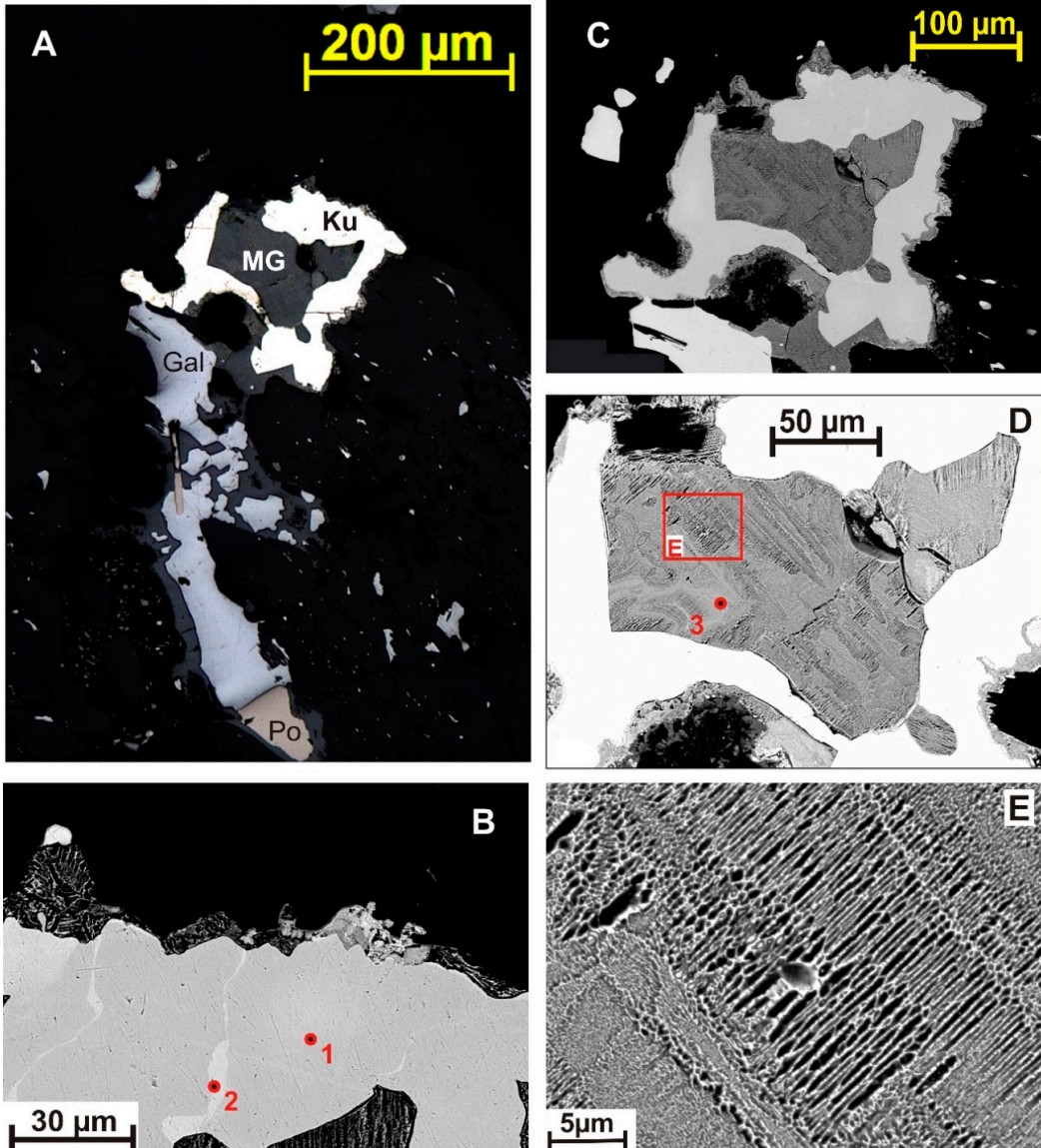

**Figure 2.** Mustard gold with küstelite. (**A**) Polished section of the mineralized quartz metasomatic rock under plane polarized light: Ku—küstelite; Gal—galena; MG—mustard gold; Po—pyrrhotite; (**B**–**E**) back scattered electron (BSE) images. (**B**) Heterogeneity of küstelite with channels of migration of electrum across küstelite; (**C**–**E**) mustard gold grain, images at different scales. Red figures (and also in Figures 3–8) are the points of analyses presented in Tables 1 and 2.

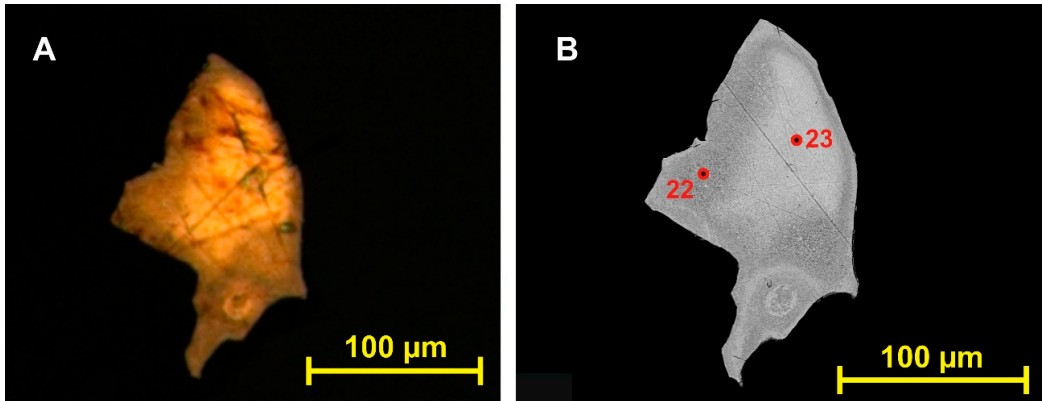

**Figure 3.** A grain of mustard gold with thallium. (**A**) Polished section under plane polarized light; (**B**) BSE-image with the points of microprobe analysis.

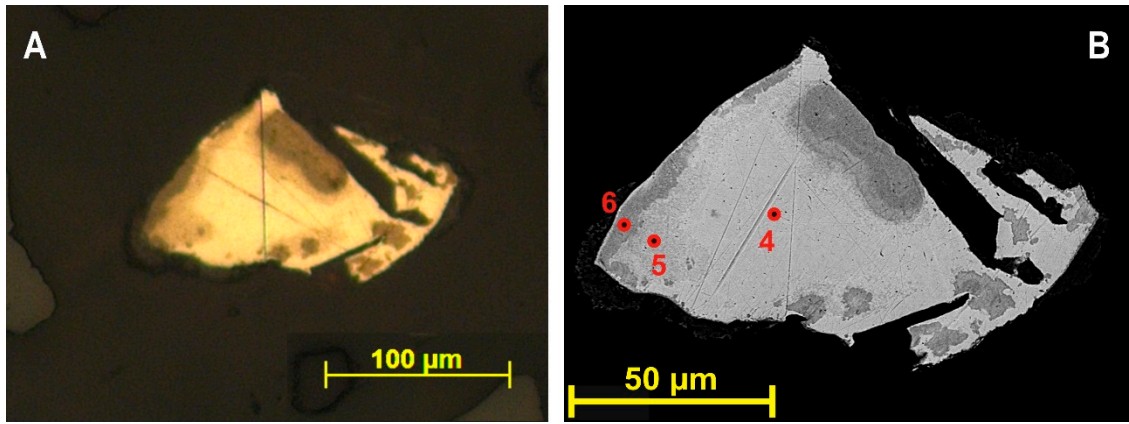

**Figure 4.** A grain of küstelite, partly replaced by mustard gold with silver chlorides and bromides. (**A**) Polished section under plane polarized light; (**B**) BSE-image of the same grains. The darkest parts of the mustard gold grains contain >3 wt% Fe.

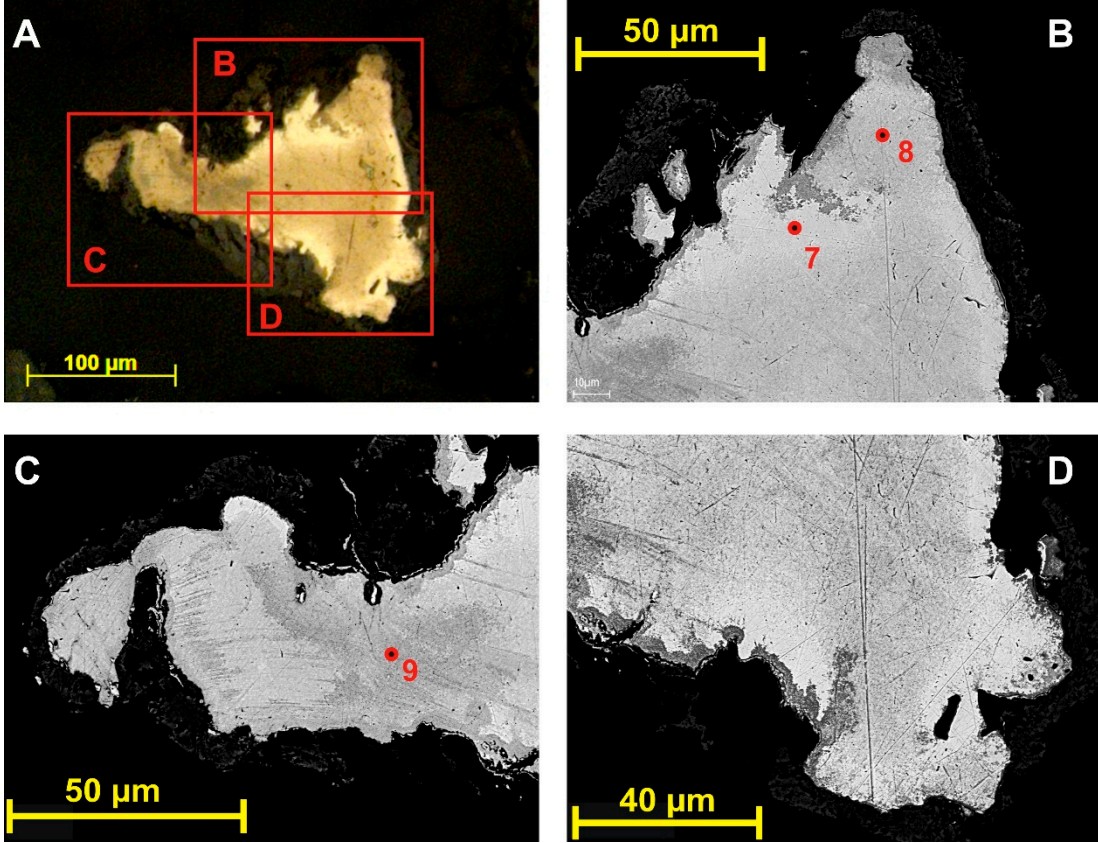

**Figure 5.** A grain of küstelite, partly replaced by mustard gold with silver chlorides and bromides. (**A**) Polished section under plane polarized light; (**B**–**D**) BSE-images of different parts of the grain, illustrating fine intergrowths of different mineral phases.

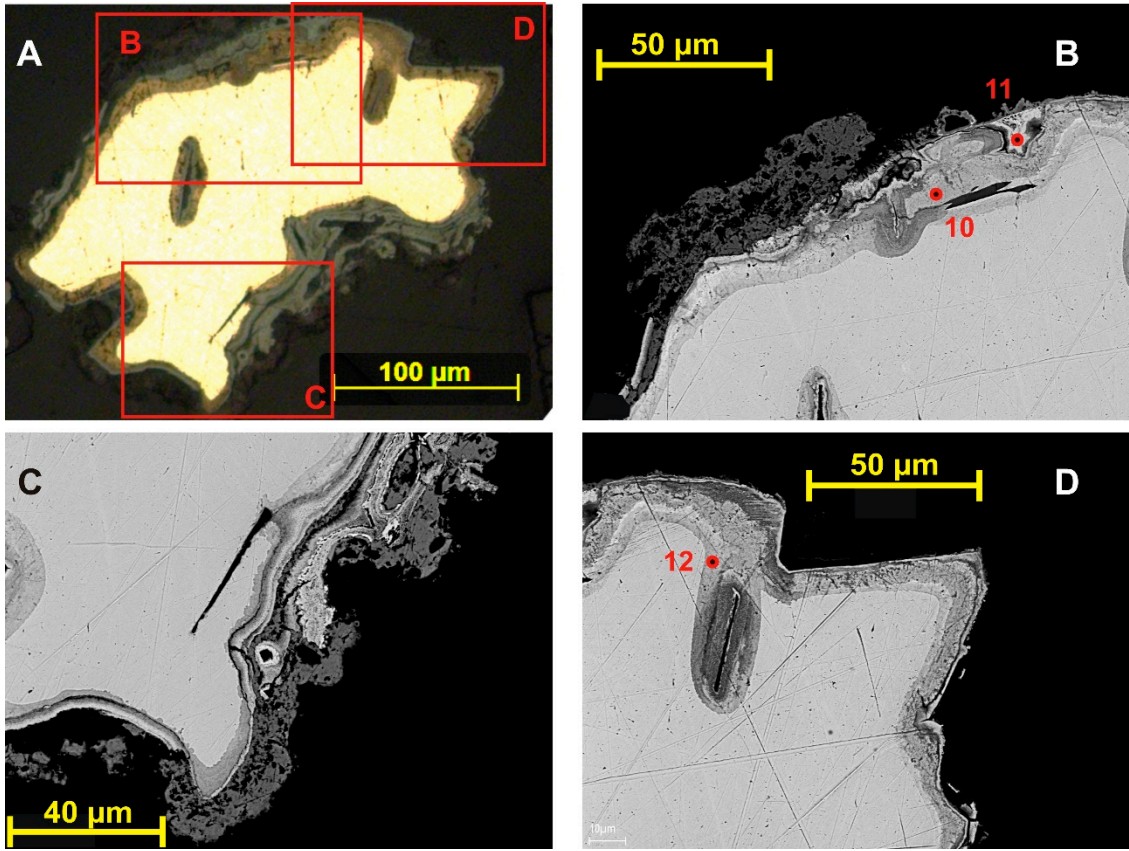

**Figure 6.** A grain of küstelite, with a rim of mustard gold with silver chlorides, bromides, and sulfides. (**A**) Polished section under plane polarized light; (**B**–**D**) BSE-images of different parts of the grain. Spongy dark-gray mineral, overgrowing the grain, is chlorargyrite.

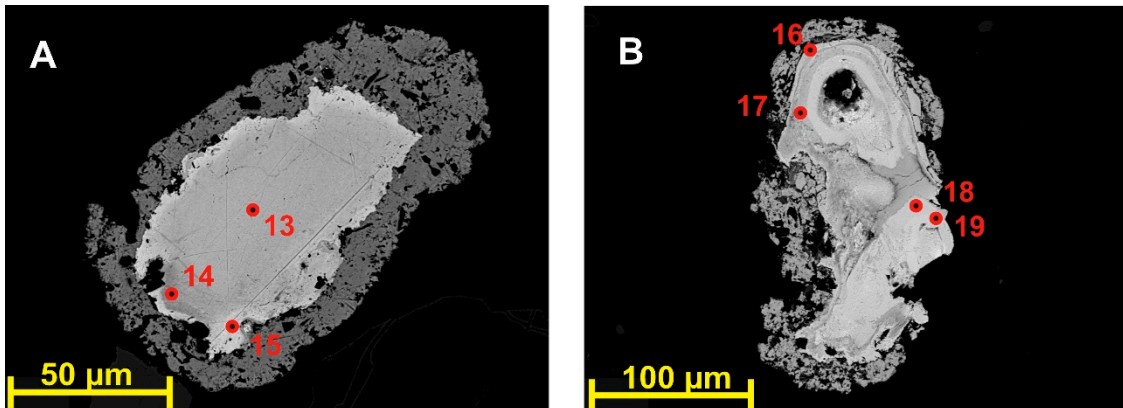

**Figure 7.** (**A**) Mustard gold with Br, Cl, Sb, C, and S in different ratios; the grain is overgrown with chlorargyrite; BSE-image; (**B**) zoning in the grain of mustard gold, connected with zonal distribution of Au and Ag, Cl and Br, halogens and S; the grain is overgrown with chlorargyrite; BSE-image.

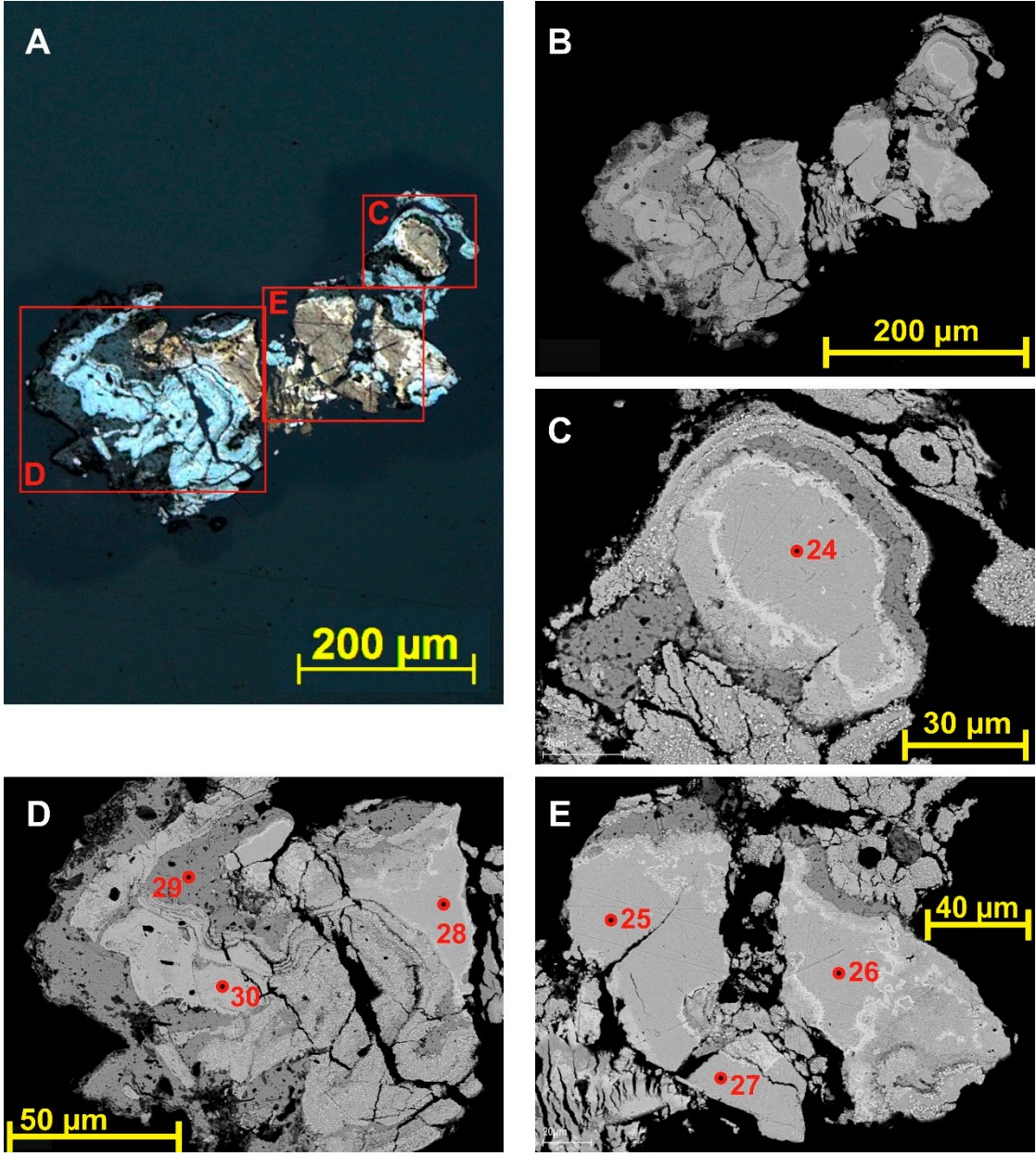

**Figure 8.** Zonal grain of mustard gold with gold and silver chlorides and sulfides. (**A**) Polished section under plane polarized light; (**B–E**) BSE-images of different parts of the grain with points of analysis.

Heavy mineral concentrates from crushed mineralized samples and from the glacial deposits overlaying the mineralized rocks contain küstelite grains, which are partly altered to mustard gold with S, Cl, and Br in its composition. The "primary" küstelite contains ~25 wt% Au, ~65 wt% Ag, and ~2 wt% Sb (Table 1, #4 and #7). Altered parts of the grains are of yellow-brown color, with dull luster (Figures 4–6). Contents of Au increase, Ag decrease, and Br, Cl, S, Fe, As (and C in one case) appear in different proportions in the altered parts of the grains (Table 1, #4–12). The substance is heterogeneous, and we see in the BSE-images (Figures 4 and 5) that it is a fine intergrowth of Au–Ag alloy with other phases.

Many grains of mustard gold are zonal (Figures 4–7) due to zoning in the distribution of Ag and Au, Br and Cl, halogens, and S. Transition from one zone to another is more often sharp, but in some grains it is gradual. Zones with chlorides and bromides of Ag and Au are yellow–brown, and those rich in S are blue-gray. Some grains are coated with chlorargyrite or goethite.

Some grains do not preserve relics of "primary" küstelite, and their cores consist of creamy-colored material (Figures 7B and 8), containing Au, Ag, Cl, and rarely, some Br and S (Table 1, #17, Table 2, #24–#28). This mustard gold is a fine intergrowth of Ag–Au alloy and chlorargyrite, the presence of both phases was verified with X-ray analysis. Conversion of the results of microprobe analysis from weight percent to atomic quantities (Table 1, #18, Table 2, #24, #25) shows that (Ag + Au)/Cl ratio is close to 2, i.e., the content of Ag–Au alloy and chlorargyrite in the intergrowth is near equal.

Mustard gold grains (intergrowths of Ag–Au alloy and chlorargyrite) are fringed with a strip, enriched in very fine (<1 μm) re-deposited electrum grains (Figure 8A,C,E).

The grains of mustard gold are surrounded by a zone of silver-rich minerals (gray in Figure 8A). This zone is of banded texture, up to 0.2 mm thick. It consists of strips of gray material with a dull luster and light blue-gray with metallic luster (Figure 8A,D). The gray, with the dull luster material contains Ag, Cl, and a little (1.2 wt%) Au, (Ag + Au)/Cl ratio is 2 (Table 2, #29). This is a very fine intergrowth of metallic Ag and chlorargyrite in equal quantities. The presence of both mineral phases was confirmed with X-ray analysis.

The composition of the light blue-gray with metallic luster mineral (Table 2, #30) corresponds to formulae $Ag_{3.2}Au_{0.8}S_2$, i.e., this is probably silver–gold sulfide yutenbogaardtite.

Chlorargyrite makes the outermost zone of loose material with a dark-gray color (better seen in Figure 7B). The thickness of this zone is up to 20 μm.

## 5. Discussion

The mustard gold forms in a process of decomposition of Au–Te and Au–Sb compounds [6,8,10,11] mainly in the zone of oxidation and disintegration of Au–Te and Au–Sb ore, and in residual placers, not actively reworked by alluvial processes [3]. Findings of mustard gold are numerous in the zones of hypergenesis of epithermal low-depth Au–Ag–Te and Au–Ag–Sb deposits in the Mesozoic and Cenozoic volcanic belts [2–14], which got to the surface and are actively denudated at present.

The Oleninskoe gold deposit differs from all other gold deposits in the Fennoscandian Shield in its geochemical and mineralogical characteristics: It is rich in Sb-sulfosalts of silver, lead, and copper, contains Au–Sb minerals gold-bearing dyscrasite and aurostibite, gold is low-grade (küstelite makes three of four generations of Ag–Au alloys), and contains 1–2 wt% Sb [22]. These specific features of the Oleninskoe deposit, similar to those in Au–Ag–Sb epithermal deposits, favored the formation of mustard gold after küstelite and Au–Ag–Sb minerals in the zone of ore oxidation.

A hydrothermal origin of mustard gold was considered to be possible along with hypergene genesis after the findings of mustard gold in the deep horizons of Au–Sb deposits in Sakha–Yakutia in aggregates with non-altered arsenopyrite, pyrite, chalcostibite, and fahlore [2,5]. The possibility of the hydrothermal genesis of mustard gold is confirmed by the results of the experiment with the decomposition of gold telluride (calaverite) in fluids with alkaline and normal pH, and the formation of mustard gold in the temperature interval 140–220 °C [25].

All samples with mustard gold in the Oleninskoe deposit were taken from the surface of bedrocks in the trenches or from the bottom of glacial deposits, overlaying the mineralized rocks. This indicates that the mustard gold formed in the zone of hypergenesis. Only one mustard gold grain in the aggregate with küstelite, non-altered galena, and pyrrhotite (see Figure 2) looks like a mineral of hydrothermal genesis.

Mustard gold is usually a product of decomposition of gold tellurides and antimonides, but it can form after Ag-rich Au–Ag alloys as well [2,3,5]. Replacement of Ag–Au alloys with impurities of Sb by mustard gold was described in one of Au–Sb epithermal deposit in Sakha–Yakutia [5]. In the Oleninskoe deposit, Ag-rich Au–Ag alloy küstelite with 1–3 wt% of Sb is the main mineral, which is replaced by mustard gold. Another mineral, which is supposed to produce mustard gold, is gold-bearing dyscrasite (up to 17 wt% of Au). Many dyscrasite grains in the Oleninskoe deposit have the porous structure of decomposition of the mineral (Figure 9), very similar to that one in the mustard gold.

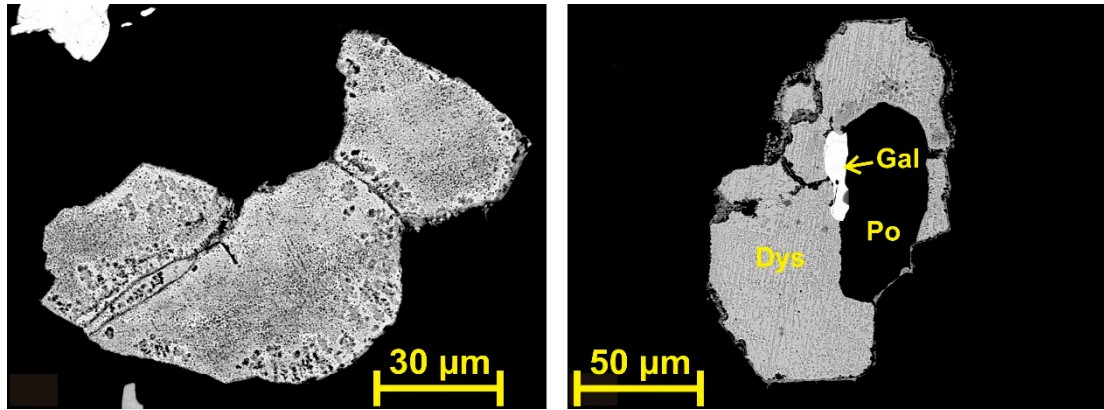

**Figure 9.** Altered Au-bearing dyscrasite with a porous structure, BSE-images. Po—pyrrhotite, Gal—galena, Dys—dyscrasite.

As is shown in [26,27], silver is more mobile than gold in low-temperature processes, and acid or neutral fluids can easily mobilize silver in complexes with $Cl^-$ or $Br^-$ from the Ag–Au alloys. This process is realized during the process of the formation of mustard gold in the Oleninskoe deposit. A "gold sponge" in the mustard gold grains forms through mobilization and partial removal of Ag with a corresponding increase in gold content in the residual material: The composition of the "primary" alloy corresponds mainly to küstelite with Au/Ag ratio ~1/3, but in the mustard gold Au/Ag ratio is more than 1, i.e., close to that one in electrum. The mobilized silver is partially removed and can be re-deposited in the outer parts of the mustard gold grains in the form of chloride and sulfide minerals.

The substance filling the pores in the "gold sponge", is of different composition. As a rule, it contains some Fe oxides (or hydroxides)—probably it is goethite with impurities of As, Sb, Al, Si. Generally, Fe-oxide/hydroxide is the most common substance to fill the pores [2,9], and it determines the brown color of the mustard gold grains. Except for the Fe oxide, we found high content of Sb (>17 wt%) in one of the studied grains. This gave us a reason to suppose that the grain was a product of decomposition of Au-bearing dyscrasite, not küstelite.

Thallium is not mentioned in the list of elements, typical for mustard gold [8,9], and it is the first finding of mustard gold with Tl. Thallium was reported earlier as an impurity in late sulfides and oxides, which form thin (1–3 μm) rims around Ag–Au grains and, rarely, around pyrite and arsenopyrite [28] in the Oleninskoe deposit. Thallium-bearing minerals were crystallized, by all indications, at the very late stages of hydrothermal ore alteration [28].

The mustard gold with silver chlorides and bromides filling the pores has not been described earlier. In the zone of ore oxidation in the Oleninskoe deposit, we can see the gradual replacement of küstelite by mustard gold with the formation of zonal grains. Silver, mobilized from küstelite during its decomposition, filled the pores in the form of chlorargyrite and bromargyrite, and it was partially removed and re-deposited in the outer parts of the mustard gold grains.

Silver is a metal less stable than gold under the conditions of ore oxidation [26,27,29]. Hence, it hardly can form structures similar to mustard gold. Nevertheless, native silver can form aggregates of very thin filaments, as was described for the process of removal of extra silver from freibergite [30]. Fine intergrowth of native silver with chlorargyrite, structurally similar to mustard gold, was found in the Oleninskoe (Figure 8, outer zone of the grain). But mechanism of formation of this intergrowth differs from that one of mustard gold: The native silver–chlorargyrite intergrowth formed in the outer parts of the mustard gold grains of the elements, mobilized from the "primary" küstelite, whereas the mustard gold formed in situ of the residual substance of the "primary" grain.

## 6. Conclusions

Mustard gold is a rare mineral formation, and finding it in the Oleninskoe deposit is the first finding in the Precambrian Fennoscandian Shield. In the Oleninskoe deposit, the mustard gold formed

as a product of decomposition of the Ag–Au–Sb minerals—küstelite with an impurity of Sb and, probably, gold-bearing dyscrasite.

The studied mustard gold is composed of a net of very fine gold filaments ("gold sponge") with pores filled with Fe and Sb oxides or Ag chlorides. Mustard gold grains are non-homogenous. Some of them are a mosaic structure, connected with change in pore size and orientation of the gold filaments in different microblocks. Many mustard gold grains are zonal, and zonation formed due to partial removal of Ag and corresponding enrichment in gold of the residual material and due to uneven distribution of the substance filling the pores.

Thallium oxides, silver chlorides, bromides, and sulfides have not been mentioned earlier as a substance to fill the pore in mustard gold. Tl, Cl, Br, and S extend the list of elements found in the mustard gold.

Native silver–chlorargyrite fine intergrowths, structurally similar to mustard gold, were found in the outer parts of the mustard gold grains. These intergrowths probably formed of the elements, mobilized from altered küstelite during the formation of mustard gold.

**Author Contributions:** Methodology, A.A.K.; Conceptualization, A.A.K.; Original Draft Writing, A.A.K.; Electron Microscope Study and Microprobe Analysis, Y.E.S.; X-ray Analyses, E.A.S.; Editing and Review, A.A.K., Y.E.S., E.A.S.

**Funding:** The work was carried out under Project 0226-2019-0053 of the Russian Academy of Sciences.

**Acknowledgments:** The authors thank Valentina Basalaeva, Anna Kalachyova, and Mikhail Sidorov (GI KSC RAS) for the preparation of heavy mineral concentrates and polished sections. The authors are grateful to the reviewers for their comments, which helped to improve the manuscript.

**Conflicts of Interest:** The authors declare no conflict of interest.

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
