# Peer review of "Mustard Gold in the Oleninskoe Gold Deposit, Kolmozero–Voronya Greenstone Belt, Kola Peninsula, Russia"

_minerals, doi:10.3390/min9120786_

Round 1

Reviewer 1 Report

The manuscript describes an interesting occurrence of gold in a precambrian terrane. This will be suitable for publication in minerals after further revisions and review, subject to the Editor's decision.

Please see my detailed comments and suggestions, regarding the structure of the manuscript and further clarification on terminology used in the text. 

Figures are of a high quality, and I believe no changes are needed to your scientific approach.

The inclusion of the experimental work in the introduction is confusing and a bit misleading, as for a moment, the reader begins to think this paper is an experimental study, or that it is a part of the current work. This material should be in the discussion to support your arguments there.

The manuscript uses inappropriate language "stuff", etc. this need to be taken out and replaced with proper scientific terminology.

There is confusion over the use of the term "cell". Needs to be better defined and constrained.

Stick with at. % or wt.% in the description of concentrations. 

Grades should be in g/t or Oz/t.

The authors need to state the history of usage of the term 'mustard Gold' with past references.

This Manuscript can be published in Minerals, but will need significant editing and some restructuring.

Interesting work.

Kind Regards,

Author Response

Dear reviewer,

Thank you for the notes to the article, they were very helpful, and we tried to correct all of them.

Special thanks for English editing!

In the table below we give some replies to your notes.

Reviewer’s note

Authors’ reply

The inclusion of the experimental work in the introduction is confusing and a bit misleading, as for a moment, the reader begins to think this paper is an experimental study, or that it is a part of the current work. This material should be in the discussion to support your arguments there.

The manuscript uses inappropriate language "stuff", etc. this need to be taken out and replaced with proper scientific terminology.

There is confusion over the use of the term "cell". Needs to be better defined and constrained.

Stick with at. % or wt.% in the description of concentrations. 

Grades should be in g/t or Oz/t.

The authors need to state the history of usage of the term 'mustard Gold' with past references.

This material is shortened and moved to ‘Discussion’

Replaced with ‘substance’ or ‘material’

Replaced with ‘pore’

Corrected

Corrected

Partly done. This special issue of Minerals on ‘Mineralogy of noble metals…’ contains one more article about mustard gold (by N.Tolstykh, G.Palyanova et al.), where these historical details are considered. That’s why we did not put much attention to the history of usage of the term ‘mustard gold’

Arkadii Kalinin

Reviewer 2 Report

The article discusses the results of the study of mustard gold in the Oleninskoe gold deposit (Kola Peninsula). Mustard gold was detected in some Meso-Cenozoic epithermal deposits, as well as in hypergenic environments. Its formation is associated with the decomposition of gold tellurides, as well as antimonides and Au-Ag alloys. The specificity of this study is that mustard gold was found in ores of the Precambrian deposit on the Baltic shield.

The article can be published with some comments.

Part 2.

Is there any information about age of mineralization?

Mineralization is discordant to the strike of host rocks. What is its formation connected with?

Is there any data on the conditions of formation (thermobarogeochemistry or geothermometers)?

Is there any data on the hypergenesis zone?

Part 3.

X-ray analysis used for identification of phases in mustard gold - is this a local method? What is the degree of locality?

Reflectivity is mentioned for characterizing of gold. How was it measured and why for only one type of gold?

Part 4.

The section is not very well structured. As I understand, the authors consider the Oleninsky deposit to be epithermal and mustard gold is the primary mineral.

There are no sufficient reasons to attribute the Oleninsky deposit to the epithermal type. The conclusions that mustard gold formed in the hydrothermal stage are not entirely justified. On the other hand, the association with limonite, silver halogenides, is typical for hypergenesis.   

I hope that these considerations and comments will prove useful

Author Response

Dear reviewer,

Thank you for your comments to the manuscript, we tried to correct the text according to your notes.

Part 2.

Is there any information about age of mineralization?

Mineralization is discordant to the strike of host rocks. What is its formation connected with?

Is there any data on the conditions of formation (thermobarogeochemistry or geothermometers)?

Is there any data on the hypergenesis zone?

Reply: We did not include much information about geology of the Oleninskoe deposit because the article is concentrated mainly on mineralogical characteristics. Now these data are added to the text, but without details, with references to published papers.

Part 3.

X-ray analysis used for identification of phases in mustard gold - is this a local method? What is the degree of locality?

Reflectivity is mentioned for characterizing of gold. How was it measured and why for only one type of gold?

Reply: Information about X-ray analysis is added to the text. Data on reflectivity are deleted from the article, because the reflectivity was measured 35 years ago, and now it is impossible to reconstruct the regime of measuring process.

Part 4.

The section is not very well structured. As I understand, the authors consider the Oleninsky deposit to be epithermal and mustard gold is the primary mineral.

There are no sufficient reasons to attribute the Oleninsky deposit to the epithermal type. The conclusions that mustard gold formed in the hydrothermal stage are not entirely justified. On the other hand, the association with limonite, silver halogenides, is typical for hypergenesis.

Reply: No, we do not consider the Oleninskoe deposit as an epithermal one (probably it can be classified as an intrusion related), but we see it has some mineralogical characteristics similar to epithermal deposits, and these features favor formation of the mustard gold. Mustard gold formed mainly in the zone of hypergenesis, but maybe some grains at the hydrothermal stage.

We changed the structure of the chapter and tried to explain our point of view more clearly.

Thank you,

Arkadii kalinin